# Characteristics of Patients Treated with JAK Inhibitors in Rheumatoid Arthritis before versus after VTE Risk Warnings

**DOI:** 10.3390/jcm12010207

**Published:** 2022-12-27

**Authors:** Cécile Philippoteaux, Valentine Deprez, Aurore Nottez, Emeline Cailliau, Eric Houvenagel, Xavier Deprez, Peggy Philippe, Tristan Pascart, René-Marc Flipo, Vincent Goëb, Jean-Guillaume Letarouilly

**Affiliations:** 1Department of Rheumatology, Lille University Hospital, 59000 Lille, France; 2Department of Rheumatology, Amiens University Hospital, 80000 Amiens, France; 3Department of Rheumatology, Dunkerque Hospital, 59240 Dunkerque, France; 4Department of Biostatistics, Lille University Hospital, 59000 Lille, France; 5ULR 2694—METRICS: Evaluation of Health Technologies and Medical Practices, Lille University, 59000 Lille, France; 6Department of Rheumatology, Catholic Saint Philibert Hospital, 59160 Lomme, France; 7Department of Rheumatology, Valenciennes Hospital, 59300 Valenciennes, France

**Keywords:** rheumatoid arthritis, JAK inhibitors, venous thromboembolic risk, tolerance

## Abstract

Background: Baricitinib (BARI) or Tofacitinib (TOFA) were the first Janus Kinase Inhibitors (JAKi) to be marketed in rheumatoid arthritis (RA). Concerns regarding venous thromboembolism (VTE) risk have emerged during the past years. The aim of the study was to compare the baseline characteristics of patients initiating BARI or TOFA in RA before versus after European Medicine Agency (EMA)’s VTE warnings and to compare real-world persistence with these two drugs. Methods: In this multicentric cohort study, RA patients initiating BARI or TOFA were included from October 2017, date of BARI marketing authorization in France, to September 2020. Baseline characteristics regarding VTE risk were compared (before vs. after May 2019) by using pre-specified statistical tests. Comparison of persistence was assessed by using propensity-score methods. Results: 232 patients were included; 155 with BARI and 77 with TOFA. Baseline characteristics of patients regarding VTE risk factors were not statistically different when Janus Kinase inhibitor (JAKi) was initiated before vs. after EMA’s warnings although a trend towards a lower proportion of VTE history was observed. Five VTE events occurred, four with BARI, one with TOFA. Cumulative persistence rate at 2 years was similar between BARI and TOFA: HR 0.96; 95% Cl: 0.52 to 1.74; *p* = 0.89. Conclusions: Our study did not show a significant change in patients characteristics starting a JAKi after the EMA’s warnings, probably due to a lack of power. Though, the lower proportion of VTE history in patients after May 2019 suggests that rheumatologists have taken into account the potential VTE risk. These results need to be confirmed by further evidence.

## 1. Introduction

In recent decades, the therapeutic arsenal for rheumatoid arthritis (RA) has dramatically expanded [1,2,3]. The Janus Kinase Inhibitors (JAKi) are the first representative of the targeted synthetic disease-modifying antirheumatic drugs (tsDMARDs) in RA [4]. They interfere with the intracellular JAK/STAT pathway implicated in RA pathogenesis and progression [5]. Baricitinib (BARI) and Tofacitinib (TOFA) were the first JAK inhibitors (JAKi) to be recommended for moderate-to-severe RA after conventional synthetic disease-modifying antirheumatic drug (csDMARD) failure [6]. Several pivotal studies have proven these molecules to be efficient either in monotherapy or in combination with a csDMARD in different subgroups of RA [7,8,9,10,11,12]. Safety data from the pivotal studies and their extension showed a tolerance profile similar to that of biological disease-modifying antirheumatic drugs (bDMARDs) with the exception of an increased risk of herpes zoster [13,14]. 

However, several alerts have arisen in recent years concerning an increased risk of venous thromboembolism (VTE) with BARI and more recently with TOFA [15,16]. In May 2019, the European Medicines Agency (EMA) sent a warning about an interim post-marketing analysis revealing a significant increase in the number of VTE and deaths with TOFA at the dose of 10 mg twice daily [17]. To date, there is no evidence that JAKi used at doses recommended in RA are responsible for more venous thromboses than bDMARDs [18].

The first objective of this retrospective multicentric study was to evaluate and compare the baseline characteristics regarding VTE risk factors of patients initiating BARI or TOFA before vs. after EMA’s warning to determine if use restrictions have been applied in daily practice. The baseline characteristics at interest for our patients were the body mass index (BMI), the age, the personal history of deep venous thrombosis (DVT) or pulmonary embolism (PE), the smoking status and the neoplastic history. The aim of our paper was to evaluate if 2019′s VTE alerts have changed the profile of patients starting a JAK inhibitor in RA. The second objective of our study was to compare the real-world persistence with BARI and TOFA in order to evaluate the potential differences in terms of tolerance or efficacy between these two molecules in a real-life setting. 

## 2. Materials and Methods

### 2.1. Study Design and Participants

A retrospective, national, multicentric (5 centers) cohort study was conducted. RA Patients fulfilling the 2010 ACR/EULAR RA classification criteria [19] and initiating BARI or TOFA were included between October 2017, date of BARI marketing authorization in France, and September 2020. TOFA was approved in France in January 2018. Patients were identified in one center thanks to the International Classification of Diseases-10 diagnostic code (M06.9). One center had an electronic database on patients treated with BARI and TOFA with all the information needed for this study, already described in a published report [20]. The three other departments of Rheumatology had a registry of RA patients treated with tsDMARDs. In all cases, data were extracted from patient’s medical files. The decision of introducing a JAKi was made by the referent physician of the patient, according to the European and French Rheumatology Society recommendations [6,21], which is a moderate-to-severe RA insufficiently controlled by a csDMARD or a bDMARD. TOFA or BARI use as monotherapy or with a csDMARD was at the rheumatologist’s discretion. All patients were JAKi-naïve. Initiation of BARI or TOFA was the index date. Patients were followed up to 24 months after the initiation. Demographic and clinical data at baseline were collected from patient’s medical files. 

### 2.2. Outcome Measure

The first primary end point was to compare the baseline characteristics of patients initiating BARI or TOFA before vs. after EMA’s VTE warning letter in May 2019. VTE risk factors were collected at baseline. The second primary end point was to evaluate and compare the persistence with BARI and TOFA at two years. Persistence was defined as the time from initiation to discontinuation. Subjects who did not discontinue the treatment at the end of the observation period were censored at the date of the last visit. A secondary endpoint was to evaluate if the overall persistence with JAKi was affected by the combination with a csDMARD. This study also aimed to explore if the line of the JAKi impacted the persistence rate. Reasons of discontinuation were collected (primary inefficacy, secondary inefficacy, adverse events including VTE events, remission and other reasons such as pregnancy). Primary inefficacy was defined as an inadequate response to treatment at 6 months (i.e., increase or stable DAS 28 CRP composite disease activity score and/or stable or deterioration of the physician’s global assessment). Secondary inefficacy was defined as a loss of response to treatment after an initial response.

### 2.3. Statistical Analyses

Continuous variables are expressed as means (standard deviation) in the case of normal distribution or medians (interquartile range, IQR) otherwise. Categorical variables are expressed as numbers (percentage). Normality of distributions was checked graphically and using Shapiro–Wilk test [22]. Baseline characteristics were described according to the treatment (BARI vs. TOFA) and the magnitude of the between-group differences were assessed by calculating the absolute standardized difference (ASD); an ASD <10% was retained as a clinically non-significant difference, ensuring comparability of our two groups [23]. Cumulative persistence was estimated using the Kaplan–Meier method by considering treatment withdraw as an event of interest [24].

The baseline characteristics of patients regarding VTE risk factors were compared between the two periods (before vs. after May 2019) by using Chi-square test (or Fisher’s exact test in case of expected value <5) for categorical variables and Student’s *t*-test (or Mann–Whitney U test in case of non-Gaussian distribution) for quantitative variables.

The difference in treatment persistence between BARI and TOFA was assessed by accounting pre-specified potential confounding factors by using propensity score (PS) weighting method using overlap weights (PSOW) as primary analysis, and PS matching as secondary (sensitivity) analysis (Appendix A) [25,26]. The PS was estimated using a non-parsimonious multivariable logistic regression model, with the treatment group as the dependent variable and all of the baseline characteristics listed in Table 1 as covariates. Patients from the TOFA group were matched to the BARI group with a maximum ratio of 1:2 according to PS using the greedy nearest neighbor matching algorithm with a caliper width of 0.2 SD of logit of PS [27]. To evaluate bias reduction using the propensity score matched method, ASD were calculated in the PS-matched cohort [23]. Treatment effect size (hazard ratio of treatment withdraw for TOFA vs. BARI treatment) was estimated using a weighted Cox proportional hazard model with overlap weights for primary analysis and using a marginal Cox proportional hazard model for sensitivity analysis with a robust sandwich variance estimator to account the matched design [28,29].

Because of missing data on covariates included in PS calculation, the treatment effect sizes were estimated after handling missing values by multiple imputation using a regression switching approach (chained equations with m = 10 imputations obtained) [30]. Imputation procedure was performed under the missing at random assumption using all baseline characteristics (Table 1) with predictive mean matching method for quantitative variables and logistic regression (binary, ordinal, or polynomial) for categorical variables [31]. In each imputed dataset, we calculated the propensity score, the overlap weight and assembled a matched cohort to provide both weighted-adjusted and matched treatment effect sizes, which were later combined using the Rubin’s rules [32,33]. Further analyses were repeated among patients who received the standard posology by using both PSOW and PS matching method as done in overall study population.

The treatment persistence (both treatments combined together) was compared according to number of previous bDMARD (0 vs. 1 or 2 vs. ≥3) by using a Cox proportional hazard model [34]. 

Statistical testing was conducted at the two-tailed α-level of 0.05. Data were analysed using the SAS software version 9.4 (SAS Institute, Cary, NC, USA).

## 3. Results

### 3.1. Study Population

A total of 232 patients were included: 155 patients with BARI and 77 patients with TOFA. Women were, respectively 76.1% and 66.2% of the population in the BARI and the TOFA group. Median duration of RA was 11 years (IQR, 4 to 20) for BARI and 11 (IQR, 6 to 19) for TOFA. Combination with a csDMARD was reporter in 38.7% of patients in the BARI group and 37.4% of patients in the TOFA group. The proportion of bDMARD-naïve patients was 14.8% with BARI and 9.1% with TOFA. BARI was prescribed at the reduced posology of 2 mg per day in 29 patients (18.7%). A proportion of 69% of patients in the BARI group and 70.1% of patients in the TOFA group initiated the treatment before May 2019. A personal history of arterial hypertension was reported in 31% (BARI) and 36.4% (TOFA) of patients. Neoplastic history concerned 7.1% of patients with BARI and 7.8% of patients with TOFA. 

Table 1 presents the baseline characteristics after handling missing values by multiple imputations according to the two study groups before and after propensity score-matching (Appendix A for baseline characteristics before matching and handling missing values). Considering meaningful differences in main baseline characteristics defined as absolute standard differences >10%, the two study groups were well-balanced after propensity score-matching except for smoking status (ASD 14.3%).

### 3.2. Comparison of the Baseline Characteristics of Patients Regarding VTE Risk Factors before vs. after EMA’s Warnings

One hundred and sixty-one out of 232 (69%) patients initiated BARI or TOFA before May 2019. The baseline characteristics were not statistically different between the two periods (Table 2). Mean age was 59.1 ± 13.6 years old before May 2019 and 57.5± 14.8 years old after (*p* = 0.40). Mean BMI was 27.4 ± 6.2 before and 26 ± 5.1 after (*p* = 0.17). Thirteen out of 161 patients (8.1%) in the before group and 2/71 (2.8%) in the after group had a personal history of VTE. Of the 13 patients in the before group, 4 had joint DVT and PE, 6 had DVT alone, 3 had PE corresponding to 10 patients with a history of DVT within this group and 5 with a history of PE. Of the 2 patients in the after group, 1 had joint DVT and PE and 1 had PE alone, corresponding to 2 patients with a history of PE within this group and 1 with a history of DVT.

### 3.3. Persistence of Baricitinib and Tofacitinib

The cumulative persistence rate at 2 years was 39.3% for BARI and 42.8% for TOFA (Figure 1). In the PSOW analysis, persistence was not significantly different between the two JAKi: hazard ratio (HR) 0.96; 95% CI, 0.52 to 1.74; *p* = 0.89. Similar findings were found in the propensity score-matched analysis: HR 0.93; 95% CI, 0.59 to 1.45; *p* = 0.75 (Table 3).

#### 3.3.1. Overall Drug Persistence in Monotherapy vs. Combination with csDMARDs

The impact of combination therapy with csDMARD on overall JAKi persistence was studied. Persistence between monotherapy vs. combination with csDMARDs was not significantly different; (HR) 1.11; 95% CI: 0.75, 1.65; *p* = 0.60 (Appendix A).

#### 3.3.2. Overall Drug Persistence Depending on Previous bDMARD Status

Persistence was not statistically different between the three previous bDMARD status, *p* = 0.37. Using patients with bDMARD-naïve as reference, the hazard ratio of treatment discontinuation was 1.78 (95%CI, 0.80 to 3.97) for 1–2 previous bDMARD(s) and 1.66 (95%CI, 0.76 to 3.64) for 3 or more previous bDMARDs (Appendix A).

### 3.4. Tolerance

Seventy-eight out of 155 patients discontinued BARI because of primary inefficacy in 24/75 patients (32%), secondary inefficacy in 24/75 patients (32%), an adverse event in 23/75 patients (30.7%) and other causes (pregnancy, lack of observance and comorbidity decompensation without apparent link with RA nor JAKi) in 4/75 patients (5.3%) (Appendix A). Adverse events leading to BARI discontinuation included gastrointestinal manifestations (diarrhea, abdominal pain, dyspepsia or vomiting) in 8/23 patients (34.8%), infection in 6/23 patients (26.1%) including herpes virus infection in 3/23 patients. 

Thirty-five out of 77 patients discontinued TOFA because of primary inefficacy in 14/35 patients (40%), secondary inefficacy in 7/35 patients (20%), an adverse event in 11/35 patients (31.4%) and other causes (lack of observance and comorbidity decompensation without apparent link with RA nor JAKi) in 3/35 patients (8.6%) (Appendix A). Adverse events leading to TOFA discontinuation included infection in 3/11 patients (27.3%) and digestive manifestations in 1/11 patients (9.1%).

### 3.5. Focus on VTE Events

Five out of 232 patients presented a VTE event during our study, 4/155 patients (2.6%) in the BARI group and 1/77 patient (1.3%) in the TOFA group. None of the events were fatal (Table 4). One out of 5 patients presented a deep venous thrombosis (DVT), 2/5 patients presented pulmonary embolism (PE) and 2/5 patients presented both DVT and PE. Delay of occurring based on the JAKi onset was 8 to 23 months with BARI and 9 months with TOFA. Mean age of occurring was 67.8 ± 13.9 years old and the 5 patients were women. Four out of 5 patients had a high BMI. Two out of 5 patients had a personal history of venous thromboembolism. All patients had at least one cardiovascular risk factor in addition to RA. Four out of 5 events occurred in patients who initiated JAKi before May 2019. Four out of 5 events lead to definitive treatment discontinuation. One event led to treatment posology reduction (BARI 4 mg per day to 2 mg per day).

## 4. Discussion

We included 155 patients with BARI and 77 patients with TOFA. The lower proportion of patients in the TOFA group can be explained by various factors: a later marketing than BARI, the constraint of administration twice daily (the prolonged-release tablet of TOFA was not available in France at the time of the study) and the EMA’s alerts on TOFA that might have slowed down practitioners regarding treatment initiation. 

In comparison with the population in the pivotal studies of BARI and TOFA in refractory RA, the mean age in our study was slightly higher and the disease duration slightly lower [9,35]. The mean age and the median disease duration in our population were similar to those of real-world registries on TOFA such as the Swiss RA registry and the US Corrona registry [36,37]. In our cohort, 12.9% of patients were bDMARD-naïve and more than half of the patients initiated a JAKi as fourth line or more. These results are similar to the findings of the US Corrona registry in which only 11% of patients were bDMARD-naïve [37]. The efficacy of JAKi in refractory RA has been proven in real-world setting [38]. Concerning the combination with a csDMARD, real-world data have suggested a majority of monotherapy use [39,40]. Our study confirms this trend with 61% of monotherapy use. Compared to TNF inhibitors, JAKi are indisputably more often prescribed as monotherapy [41]. 

The comparison of the baseline characteristics of patients who initiated BARI or TOFA before vs. after May 2019 did not show a significant difference. However, a clear trend towards a lower proportion of venous thromboembolic history in the after group was observed (1.4% after vs. 6.2% before for DVT and 2.8% after vs. 4.3% before after for PE), suggesting that use restrictions have been partially applied in current practice. Patients were discretely younger after the EMA warnings, with a shorter disease duration, but not significantly. The statistical insignificance can be explained by a lack of power of our study, with too few patients included after May 2019. JAKi use restrictions according to patient characteristics can also be challenging in refractory or difficult to treat (D2T) RA as these drugs may represent the last therapeutic option. This retrospective multicentric study provides observational evidence of how these drugs are administered in a French routine clinical practice. 

The rate of VTE events in our study (2.6% with BARI and 1,3% with TOFA) is high compared to other registries and observational studies. An American cohort study found no evidence for an increased risk of VTE for TOFA vs. TNF inhibitors [42]. Data from the US Corrona registry reported 3 thromboembolic events with TOFA (1 DVT and 2 PE) in a cohort of 558 patients and a total of 24 thromboembolic events with bDMARDs [37]. The ORAL surveillance study (NCT02092467) comparing TOFA with TNF inhibitors revealed an increased and dose-dependent number of VTE events with TOFA [43,44]. In contrast, a meta-analysis of data from clinical trials of TOFA vs. placebo revealed a reassuring VTE risk, but long-term extension data were not pooled [45]. A long-term safety analysis of BARI showed no significant increased VTE risk [46]. Thus, the question of VTE risk remains a concern in non-selected RA patients [47]. RA itself has been associated with an inherent increased thromboembolic risk [48,49,50]. Moreover, an association between VTE risk and disease-activity has been described [51], suggesting that the attribution of an increased VTE risk to JAKi is difficult to assess in refractory RA. Mechanistic understanding is lacking as contradictory data exist. For instance, selective JAK1 and JAK2 Ruxolitinib has been shown to reduce VTE events in patients with polycythaemia vera and myelofibrosis [52]. Whether an increased VTE risk is a class effect of JAKi needs to be determined in the future. Until this day, safety data of Filgotinib and Upadacitinib have not revealed an increased risk of thromboembolism [53,54]. Consequently, in the longer term, larger observational studies are needed to accurately quantify the potential venous thromboembolic risk attributable to JAKi, and differentiate these from risks attributable to RA itself and its comorbidities.

The examination of the characteristics of the five patients in whom VTE events occurred in our study revealed that they were all initially at risk. Four of them had a high BMI and two of them had a personal history of VTE. Interestingly, 4 out of 5 events occurred in patients who initiated JAKi before May 2019. It can be assumed that the decision of initiating BARI or TOFA would have not been made after May 2019, especially in the two patients with a VTE history. 

The cumulative persistence rate at 2 years was 39.3 % for BARI and 42.8 % for TOFA, without significative difference, which is consistent with the results of real-world studies [39,55,56]. The persistence with BARI and TOFA was not different in propensity-score analysis. Even though BARI is a more significant inhibitor of JAK2 and a less inhibitor of JAK 3 than TOFA, the two drugs share a similar mechanism of action suggesting very similar efficacy and tolerance profiles. Overall drug maintenance was not improved by the combination with a csDMARD, which was demonstrated in other real-world studies [36,37]. Results from a Swiss cohort within the Swiss Clinical Quality Management Registry found that a higher number of previous bDMARDs was significantly associated with drug discontinuation [55]. Our study reveals a trend of a lower discontinuation rate if the JAKi was administered as first line after csDMARD failure, even though this was not statistically significant. 

The most common reason for stopping therapy in our cohort was insufficient effectiveness for both BARI and TOFA (32% and 40%, respectively). This high rate is partially explained by a high proportion of patients with refractory RA. Intolerance was the second cause of discontinuation for TOFA and the third one for BARI. The most common adverse events leading to discontinuation were digestive disturbance and infection, which is consistent with worldwide post-marketing and real-life data [57]. The data concerning the primary endpoint of the ORAL surveillance study (NCT02092467) revealed a significant cardiovascular and neoplastic increased risk with TOFA 10 mg BID compared to TNF inhibitors [44]. The STAR-RA cohort study did not demonstrate a significant cardiovascular and neoplastic increased risk with TOFA compared to TNF inhibitors, even in patients over 50 years of age with at least one cardiovascular risk factor [58,59]. No incident neoplasia nor myocardial infarction was reported in our study, but the duration of JAKi exposure was short. 

EMA’s safety committee (Pharmacovigilance Risk Assessment Committee) has recently recommended measures to minimize the risk of serious side effects associated with JAK inhibitors including cardiovascular conditions, VTE history, cancer and serious infections [60]. The Committee has recommended that these drugs should be used only if no suitable treatment alternatives are available in patients aged 65 years or above, those at increased risk of major cardiovascular problems, those a history of active or quit smoking and those at increased risk of cancer. Caution is also needed in patients with VTE risk factors other than those listed above. 

The results of our study remind us of the importance of real-world data in studying the tolerance of treatments. Unlike clinical studies, real-world patients are unselected and represent a wide range of patients, generally older with more comorbidities. 

This study has several limitations, including the inherent limitations of retrospective observational studies. Missing data and incomplete follow-up are an issue even though drug persistence is a robust outcome in this setting. We used statistical imputations due to missing data. A propensity-score weighting was used to minimize the bias that may occur with real-world data because of background characteristics. Our cohort is limited in size and the results cannot be transposed to the general population. 

## 5. Conclusions

Our study did not show a significant change in patient’s characteristics starting a JAKi after the EMA’s warnings, probably due to a lack of power. Yet, the results suggest that rheumatologists have considered the potential VTE risk, with a lower proportion of VTE history in patients starting a JAKi after May 2019. These results need to be confirmed by further evidence. BARI and TOFA have a similar real-world persistence in our study. The tolerance profile is consistent with post-marketing surveillance data and real-world registries. Our study revealed a high number of thromboembolic manifestations with a majority occurring in patients who initiated BARI or TOFA before EMA’s VTE warnings. Undoubtedly, larger observational studies are needed to accurately quantify thromboembolic risks attributable to JAKi and differentiate these from risks attributable to RA and its comorbidities. This study provides additional data and is a hot topic after the results of the ORAL-surveillance study (NCT02092467). BARI cardiovascular safety is under investigation with the ongoing RA-BRIDGE (NCT3915964) and RA-BRANCH (NCT04086745) studies. 

This study also raises the question of the place of JAKi in the therapeutic arsenal of RA. EMA’s safety Committee (PRAC) has recently confirmed that these drugs should be used only if no suitable treatment alternatives are available in at-risk patients. The rheumatologist has a key role in determining the right treatment for the right patient. 

Whether the VTE risk is a class effect of JAKi still needs to be determined. The long-term surveillance of new marketed JAKi will help us to understand whether there are specific efficacy and safety profiles among the different JAKi depending on the selectively targeted Janus kinases.

## Figures and Tables

**Figure 1 jcm-12-00207-f001:**
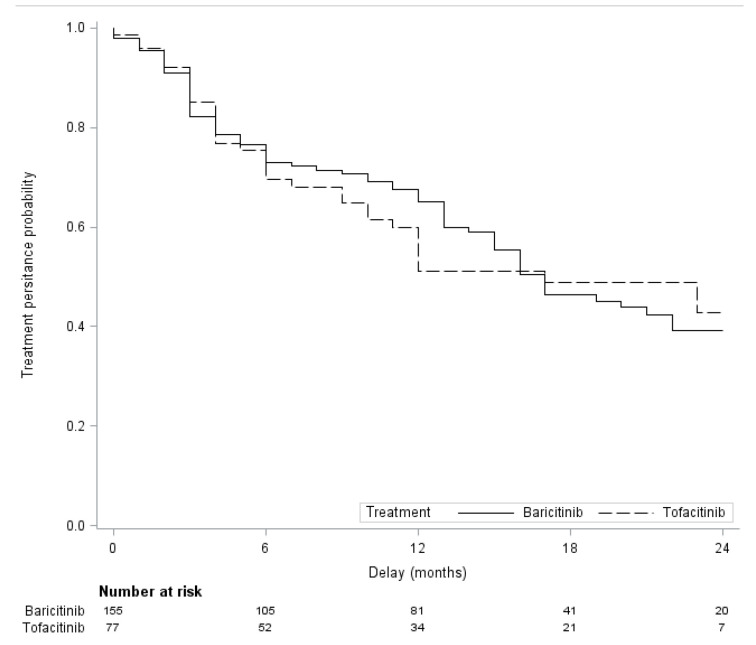
Two-year follow-up treatment persistence with Baricitinib and Tofacitinib.

**Table 1 jcm-12-00207-t001:** Baseline characteristics according to molecules before and after propensity score-matching.

	Before Propensity Score-Matching			After Propensity Score-Matching		
Parameters	Baricitinib *n* = 155	Tofacitinib *n* = 77	ASD (%)	Baricitinib *n* = 116	Tofacitinib *n* = 70	ASD (%)
Women	118 (76.1)	51 (66.2)	21.7	82 (70.8)	49 (69.6)	2.7
Age (years), mean ± SD	59.6 ± 14.2	56.7 ± 13.4	20.9	58.7 ± 15.3	57.6 ± 13.1	7.8
BMI (kg/m^2^), mean ± SD	27.0 ± 6.3	27.0 ± 6.8	4.2	26.8 ± 6.6	27.0 ± 6.5	3.0
Smoking status			31.5			14.3
Non-smoker	94 (60.7)	33 (43.4)		60 (51.2)	32 (46.1)	
Former smoker	31 (20.1)	19 (24.9)		28 (23.9)	18 (24.9)	
Current smoker	30 (19.2)	24 (31.7)		28 (24.5)	20 (29.0)	
RA duration (years), median (IQR)	11 (4 to 20)	11 (6 to 19)	10.1 *	12 (6 to 20)	12 (5 to 19)	3.0 *
Seropositivity status			21.7			8.5
RF +/ACPA +	123 (79.4)	55 (70.6)		89 (76.6)	53 (75.6)	
RF +/ACPA −	11 (7.1)	8 (10.6)		10 (8.3)	7 (9.3)	
RF −/ACPA +	8 (5.2)	7 (9.5)		7 (6.4)	4 (6.3)	
RF −/ACPA −	13 (8.4)	7 (9.2)		10 (8.8)	6 (8.9)	
Erosion	106 (68.6)	58 (75.7)	15.7	86 (74.3)	53 (75.7)	3.3
Baseline CRP (mg/L), median (IQR)	7.0 (1.1 to 25.6)	9.9 (2.6 to 27.0)	12.5 *	7.7 (1.8 to 26.8)	10.0 (2.7 to 27.2)	4.0 *
Prior bDMARD			27.7			7.8
Naïve (0)	23 (14.8)	7 (9.1)		12 (10.7)	7 (10.0)	
1 or 2	50 (32.3)	34 (44.2)		45 (38.8)	29 (42.0)	
3 or more	82 (52.9)	36 (46.8)		59 (50.6)	34 (48.0)	
Concomitant csDMARD	62 (38.7)	29 (37.4)	5.5	43 (37.1)	27 (38.7)	3.4

Values are numbers (%) unless otherwise stated. Values were calculated after handling missing data using a multiple imputation procedure (m = 10). * ASD calculated on log transformed data. Abbreviations: ACPA = anti-citrullinated protein antibodies; ASD = absolute standardized difference; bDMARD = biological disease-modifying antirheumatic drug; BMI = body mass index; CRP = C-reactive protein; csDMARD = conventional synthetic disease-modifying antirheumatic drug; IQR = interquartile range; RA = Rheumatoid arthritis; RF = rheumatoid factor; SD = standard deviation.

**Table 2 jcm-12-00207-t002:** Comparison of the baseline characteristics of the patients according to JAKi initiation before vs. after May 2019.

	Before May 2019 *n* = 161	After May 2019 *n* = 71	*p*-Value
Age (years), mean ± SD	59.1 ± 13.6	57.5 ± 14.8	0.40
BMI (kg/m^2^), mean ± SD	27.4 ± 6.2	26.0 ± 5.1	0.17
Smoking status			0.65
Non-smoker	62/110 (56.4)	22/43 (51.2)	
Former smoker	25/110 (22.7)	9/43 (20.9)	
Current smoker	23/110 (20.9)	12/43 (27.9)	
Personal history of DVT	10/161 (6.2)	1/71 (1.4)	0.18
Personal history of PE	7/161 (4.3)	2/71 (2.8)	0.73
Neoplastic history	12/161 (7.5)	5/71 (7.0)	0.91

Values are no./total no. (%) unless otherwise stated. 1 71 missing values (*n* = 42 vs. 29). Abbreviations: BMI = body mass index; DVT = deep venous thrombosis; PE = pulmonary embolism; SD = standard deviation.

**Table 3 jcm-12-00207-t003:** Hazard ratios of treatment discontinuation for patients treated with Tofacitinib vs. patients treated with Baricitinib.

	*n*	Baricitinib Rate, %	*n*	Tofacitinib Rate,%	HR (95%CI)	*p*-Value
Unadjusted analysis	155	60.7	77	57.2	1.02 (0.68 to 1.53)	0.93
PSOW analysis	155	60.9	77	56.4	0.96 (0.52 to 1.74)	0.89
PS-matched analysis	116	61.8	70	56.1	0.93 (0.59 to 1.45)	0.75

Rates indicate the cumulative incidence of treatment discontinuation at 2 years of follow-up (calculated from imputed datasets for PSOW and PS-matched analyses after applying a log transformation for treatment survival estimates). Hazard ratios are calculated for patients treated with Tofacitinib vs. those treated with Baricitinib. Abbreviations: CI = confidence interval; HR = hazard ratio; PS = propensity score; PSOW = propensity score weighting using overlap weights.

**Table 4 jcm-12-00207-t004:** Venous thromboembolic complications in our patients.

Gender/Age	RA Characteristics	Molecule/Dose/Line JAK	Comorbidities	MTEV Type	Delay of Occurring	Management Death	Treatment Discontinuation
Patient 1 F. 77 yo	9 years of evolution RF+/ACPA+ Introduction BEFORE May 2019	BARI 4 mg/day Monotherapy + prednisone 10 mg/day 4th line or more	BMI 35.2 Former smoker	PE	23 months	Ambulatory DOAC No death	No Dose reduction (BARI 4 to BARI 2 mg) Discontinuation 6 months later due to secondary inefficacy
Patient 2 F. 72 yo	Unknown years of evolution RF+/ACPA+ Introduction BEFORE May 2019	BARI 4 mg/day Monotherapy + prednisone 10 mg/day 4th line or more +	BMI 35.6 Smoking status unknown AHT FA (DOAC) Stroke 2 DVT (oral contraception)	PE + DVT	13 months	Hospitalization IV anticoagulant then DOAC No death	Yes
Patient 3 F. 82 yo	34 years of evolution RF+/ACPA+ Introduction BEFORE May 2019	BARI 2 mg/day Monotherapy + prednisone 5 mg/day 4th line or more	BMI 22.2 Smoking status unknown AHT 2 EP (surgery)	PE	9 months	Hospitalization IV anticoagulant then DOAC No death	Yes
Patient 4 F. 42 yo	5 years of evolution RF+/ACPA+ Introduction AFTER May 2019	TOFA 5 mg BID Monotherapy + prednisone (unknown posology) 4th line or more	Unknown BMI Former smoker AHT	DVT	9 months	Ambulatory No information available about AC treatment No death	Yes
Patient 5 F. 66 yo	15 years of evolution RF+/ACPA- Introduction BEFORE May 2019	BARI 4 mg/day Monotherapy + prednisone 5 mg/day 4th line or more	BMI 31.3 Never smoked	DVT + PE	20 months	Ambulatory DOAC No death	Yes

Abbreviations: F = Female; yo = years old; DVT = deep venous thrombosis; PE = pulmonary embolism; AHT = arterial hypertension; DOAC = direct oral anticoagulant; BMI = body mass index; AF = atrial fibrillation; AC = anticoagulant.

## Data Availability

No new data were created or analyzed in this study. Data sharing is not applicable to this article.

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
