# Peer review of "Characteristics of Patients Treated with JAK Inhibitors in Rheumatoid Arthritis before versus after VTE Risk Warnings"

_jcm, 2022, doi:10.3390/jcm12010207_

Round 1
Reviewer 1 Report
The authors present the JAKi first inhibitors treatment BARI and TOFA in RA patients.Is is a multicentric study with good criteria for inclusion.
They follow the side effcets is this medication and the percentage of side effcts is high for DVT.That mean is not suitable to use by the patients are more studies are need.
It is not specify if the patient they follow some rehabilitation activities and what is happen with the patients who should stop the treatment and which other medication need to use.The sclae to evaluate the clinical score according to the treatment is not specifiy.
Author Response
We thank you for your reviewing.
We did not study if patients underwent rehabilitation activities.
Because of missing data regarding activity scores, we did not include this part in our study.
As you can see above, the mean DAS 28 CRP at baseline was 4.4 +/- 1.3 for 151 patients (81 patients with missing data); the mean DAS 28 CRP at 6 months was 3.0 +/- 1.6 for 83 patients (149 patients with missing data).
Because of missing data, we focused our paper more on the tolerance of the JAKi than on the efficacy itself. Drug survival study was a way to study the efficacy of the treatments. Nevertheless, we observed a reduction in disease activity scores during time, even though we did not include these data in our final paper.
|
Baseline_DAS28_CRP |
N |
151 |
|
|
Missing data |
84 |
|
|
Mean +/- SD |
4.4 +/- 1.3 |
|
M6_DAS28_CRP |
N |
83 |
|
|
Missing data |
149 |
|
|
Mean +/- SD |
3.0 +/- 1.6 |
Reviewer 2 Report
Dear Authors,
I've read with great interest your work titled Characteristics of patients treated with JAK inhibitors in rheumatoid arthritis before versus after VTE risk warnings. Overall, I have some concerns and recommendations for providing clear information. Particularly, I listed the following comments in detail here.
In the abstract, the author needs to mention the aim of the study at the end of the background. Also, the finding of the assay could be added step by step based on methods. All of the names and terms should be completely mentioned for the first time, for example, EMA, and so on.
In the introduction, some sentences lack reference, for example, “In recent decades, the therapeutic arsenal for rheumatoid arthritis (RA) has dramatically expanded. The Janus Kinase Inhibitors (JAKi) are the first representative of the targeted synthetic disease-modifying antirheumatic drugs (tsDMARDs) in RA.”, and so on. Most importantly, the sentences are un-regularly dispersed and too simple. Please add more potential specific information on evaluate and com- 53 pare the baseline characteristics of patients initiating BARI or TOFA.
In the Methods, the author needs to mention the ingredients of the methods and also add the reference to all tests.
In discussion, discuss your results before relating them to the results of other published works.
You need to expand your discussion based on your results step by step and compare them to other studies.
Precise conclusion as it’s too short in its current form. Add a significant statement that must be structured as “what was offered by authors? Do the authors have more thoughts on this field?
Author Response
I've read with great interest your work titled Characteristics of patients treated with JAK inhibitors in rheumatoid arthritis before versus after VTE risk warnings. Overall, I have some concerns and recommendations for providing clear information. Particularly, I listed the following comments in detail here. In the abstract, the author needs to mention the aim of the study at the end of the background. Also, the finding of the assay could be added step by step based on methods. All of the names and terms should be completely mentioned for the first time, for example, EMA, and so on.
We would like to thank you for your reviewing and corrections that you suggested in order to improve our paper.
For the abstract section, we applied the corrections, as suggested. This includes a reframing of the background section with the objective as the last sentence and the complete mention of European Medicine Agency as it appears for the first time.
“Background : Baricitinib (BARI) and Tofacitinib (TOFA) were the first Janus Kinase Inhibitors (JAKi) to be marketed in rheumatoid arthritis (RA). Warnings regarding venous thromboembolism (VTE) have emerged during the past years. The aim of the study was to compare the baseline characteristics of patients initiating BARI or TOFA in RA before versus after European Medicine Agency (EMA)’s VTE warnings and to compare real-world persistence with these two drugs.”
In the introduction, some sentences lack reference, for example, “In recent decades, the therapeutic arsenal for rheumatoid arthritis (RA) has dramatically expanded. The Janus Kinase Inhibitors (JAKi) are the first representative of the targeted synthetic disease-modifying antirheumatic drugs (tsDMARDs) in RA.”, and so on.
Thank you for your comments. We were able to add some references in the introduction section, as you suggested.
Most importantly, the sentences are unregularly dispersed and too simple. Please add more potential specific information on evaluate and compare the baseline characteristics of patients initiating BARI or TOFA.
We thank you for this comment. We were able to add the sentences in bold, as you can see above, in order to add more specific information on the baseline characteristics of patients.
“The first objective of this retrospective multicentric study was to evaluate and compare the baseline characteristics of patients initiating BARI or TOFA before versus after EMA’s warnings to determine if use restrictions have been applied in daily practice. The baseline characteristics at interest in our patients were the body mass index (BMI), the age, the personal history of deep venous thrombosis (DVT) or pulmonary embolism (PE), the smoking status and the neoplastic history. The aim of our paper was to evaluate if 2019’s VTE alerts have changed the profile of patients starting a JAK inhibitor in RA. The second objective of our study was to compare the real-world persistence with BARI and TOFA in order to evaluate the potential differences in terms of tolerance or efficacy between these two molecules in a real-life setting”
In the Methods, the author needs to mention the ingredients of the methods and also add the reference to all tests.
Thank you for your comments. We were able to add references to all tests in the Methods section (as you can see in the revised manuscript). We revised the manuscript with the statistician who worked on this study and the paragraph presents all the elements necessary for the reader to understand and interpret the statistical analyses.
In discussion, discuss your results before relating them to the results of other published works. You need to expand your discussion based on your results step by step and compare them to other studies.
We worked on the discussion section as you suggested. We have rebuilt the discussion step by step in order to present the results as following:
- Presentation of the main characteristics of our population (number of patients with BARI and TOFA, mean age, mean disease duration) and the discussion with main characteristics of patients included in other studies (pivotal studies of these molecules and main registries).
- Comparison of the baseline characteristics of patients who initiated BARI or TOFA before versus after May 2019 with the discussion about the absence of statistical significance.
- Discussion about the VTE risk:
- The rate of VTE events in our study and comparison with the rate found in other registries and studies. We also developed in this paragraph the most recent data on VTE alerts with the results of the ORAL surveillance study. We have discussed the difficulty to differentiate risk attributable to JAKi and differentiate it from risk attributable to RA itself as contradictory data exists.
- Presentation of the 5 VTE events in our cohort and discussion about the characteristics of the patients in whom these events occurred. Discussion with results of the literature.
- Discussion about the persistence rate of BARI and TOFA, the impact on the association of the JAKi with a csDMARD and the main reasons of discussion. These results are compared with available data from real-life studies and registries. We discussed also in this paragraph the discontinuation due to adverse events without any case of MACE or neoplastic disease in our cohort. The tolerance paragraph also includes recent EMA’s safety committee (Pharmacovigilance Risk Assessment Committee) recommendations.
- Presentation of the limitations of our work.